



# Turbulence kinetic energy dissipation rate estimated from a WindCube Doppler Lidar and the LQ7 1.3 GHz radar wind profiler in the convective boundary layer.

Hubert Luce[1] and Masanori Yabuki[1]

[1]Research Institute for Sustainable Humanosphere, Kyoto University, Kyoto, 611-0011, Japan

*Correspondence to*: Hubert Luce (luce@rish.kyoto-u.ac.jp)

**Abstract.** From 21 August to 15 September 2022, a WindCube v2 Infrared coherent Doppler Lidar (DL) supplied by EKO

Co. (Japan) was deployed at the Shigaraki MU Observatory (Japan) near the LQ7 UHF (1.357 GHz) wind profiler in routine operation. Horizontal and vertical velocity measurements from DL were reliably obtained in the [40-300] m height range with vertical and temporal resolutions of 20 m and 4 seconds, respectively. The LQ7 wind measurements are collected with range and temporal resolutions of 100 m and 59 s, respectively, and 10-min average profiles are calculated after data quality control. Reliable LQ7 Doppler data are collected from the height of 400 m. Despite the lack of overlap in the height range, we compared

the Turbulence Kinetic Energy (TKE) dissipation rate $\varepsilon$ in the daytime planetary boundary layer estimated by the two instruments. A method based on the calculation of the one-dimensional transverse line spectrum of the vertical velocity $W$ from mean $W$ time series (TS method) was applied to DL ($\varepsilon_{DL}$). The same method was also applied to 1-min LQ7 data ($\varepsilon_{LQ7}^{TS}$) to assess its performance with respect to DL despite the poorer time resolution. A more standard method based on the Doppler Spectral width (DS) was also applied to LQ7 ($\varepsilon_{LQ7}^{DS}$) from the 10-min average profiles. We tested recently proposed models of

the form $\varepsilon = \sigma^3/L$ where $\sigma$ is half the spectral width corrected for non-turbulent effects and $L$ is assumed to be a constant or a fraction of the depth $D$ of the Convective Boundary Layer (CBL). The main results are: (1) For the deepest CBLs $(\max(D) > \sim 1.0$ km) that develop under high atmospheric pressure, the time-height cross-sections of $\varepsilon_{LQ7}^{DS}$ and $\varepsilon_{DL}$ show very consistent patterns and do not show any substantial gaps in the transition region of 300-400 m when $\varepsilon_{LQ7}^{DS}$ is evaluated with $L \sim 70\ m$, which is found to be about one tenth of the average of the CBL depth ($L \sim 0.1\ D$). (2) Hourly mean $\varepsilon_{DL}$ averaged over the [100-

300] m height range is on average about twice the hourly mean $\varepsilon_{LQ7}^{TS}$ averaged over the [400-500] m height range when $D > \sim 1.0\ km$. (3) Hourly mean $\varepsilon_{DL}$ averaged over the [100-300] m height range and hourly mean $\varepsilon_{LQ7}^{DS}$ averaged over the [400-500] m height range with $L \sim 0.1\ D$ are identical on average. Consistent with the fact that $\varepsilon$ is expected to decrease slightly with height in the mixed layer, (2) and (3) imply an uncertainty as to the exact value of the $L/D$ ratio: $\sim 0.1\ D < L < \sim 0.2\ D$. We have also studied in detail the case of a shallow ($D < \sim 0.6$ km) convective boundary layer that developed under low

atmospheric pressure and cloudy conditions. Despite the fact that hourly mean $\varepsilon_{DL}$ averaged over the [100-300] m height range and hourly mean $\varepsilon_{LQ7}^{TS}$ averaged over the [400-500] m height range show more significant discrepancies, maybe due to the different properties of the shallow convection, the time-height cross-sections of $\varepsilon_{DL}$ and $\varepsilon_{LQ7}^{DS}$ show more consistent patterns and levels.






## 1. Introduction

The planetary boundary layer is the interface between the Earth's surface and the free atmosphere. It plays an essential role in exchanges of matter, energy and momentum and in the mechanisms governing atmospheric circulation from local scales to planetary scales (*Stull, 1988*). It is still the subject of many studies because of the large variety of cases related to various meteorological and surface conditions. The turbulence kinetic energy (TKE) dissipation rate $\varepsilon$ is one of the key parameters characterizing the dynamics of the convective boundary layer (CBL). Reliable and continuous measurements of this parameter covering the whole depth of the CBL are necessary to assess and improve the subgrid turbulence schemes in numerical weather forecast models. In the present work, we show the results of measurements of $\varepsilon$ in convective boundary layers (CBLs) from a Doppler lidar and a UHF wind profiler. During a period of about four weeks, from 21 August 2022 to 15 September 2022, a WindCube v2 Infrared coherent Doppler Lidar (DL) supplied by EKO Co., Japan, (left panel of Fig. 1) was deployed at Shigaraki MU observatory (Japan) for another project related to temperature and humidity profile measurements. The 1.357 GHz wind profiler WPR LQ-7 is a UHF Doppler radar (LQ7) developed by Sumitomo Corp. (*Imai et al., 2007*) (right panel of Fig. 1), routinely operating for boundary layer and lower troposphere observations. DL provided reliable measurements from a height of 40 m to 300 m at a time resolution of 4 s and LQ7 from the height of 400 m at a time resolution of 59 s. The LQ7 data are also processed to provide deliverables at a time resolution of 10 min after averaging and data quality control. These data are available at (http://www.rish.kyoto-u.ac.jp/radar-group/blr/shigaraki/data/).

Despite the lack of height range overlap, comparisons of $\varepsilon$ estimated by different methods can be made when sampling the mixed layer of the CBL (i.e. $0.2 < z/D < 0.8$, typically), as mean values of $\varepsilon$ is not expected to vary much in this region. $\varepsilon$ from DL (hereafter noted $\varepsilon_{DL}$) is estimated from one-dimensional line spectrum of the vertical velocity $W$ from mean $W$ time series (e.g. *Lothon et al., 2009; O'connor et al., 2010, Banakh et al., 2021, Gomez et al., 2021*). The high time resolution of DL should allow us to characterize the inertial subrange of Kolmogorov-Obhukov turbulence in the CBL from which $\varepsilon$ can be determined. Despite to the poorer time resolution of LQ7 (59 s), we also applied the TS method to LQ7 from 64-point $W$ time series to estimate $\varepsilon_{LQ7}^{TS}$ and to compare with $\varepsilon_{DL}$. In the past, the TS method has rarely been tested with VHF and UHF radars because of probable contaminations from (Doppler shifted-) gravity waves in the free atmosphere when the total acquisition time exceeds $\sim 10^0$ minute (e.g. *Hocking et al., 2016*). This constraint seems totally incompatible with the total duration (~1 hour ) used to apply the TS method but we nevertheless show that the TS method applied to LQ7 can give results consistent with those obtained with DL when both LQ7 and DL measurements are made in the CBL. The Doppler Spectral width (DS) method is an alternative method that relates the part of the variance of Doppler spectrum peaks due to turbulence to $\varepsilon$ using different models depending on the characteristics of the radar and the turbulence properties. It is commonly applied to clear air VHF Stratosphere-Troposphere (ST) radars and wind profilers (e.g., *Hocking et al., 2016*) and weather radars (e.g., *Doviak and Zrnic', 1993*). The DS method was applied to LQ7 to estimate $\varepsilon_{LQ7}^{DS}$ but not to DL because the full Doppler spectrum was not available during the campaign. In the present work, we used a very simple model obtained by *Luce et al. (2018)* for the VHF MU radar and assessed with LQ7 by *Luce et al. (2023a,b)* for shear-generated turbulence. The aim was to assess the performance of this simple model to convectively generated turbulence in the CBL from comparisons with DL measurements. In section 2, we describe the methods for estimating $(\varepsilon_{DL}, \varepsilon_{LQ7}^{TS})$ and $\varepsilon_{LQ7}^{DS}$ and the practical procedure used for the comparisons between the two instruments. We show the results for two case studies and statistics based on data collected during the whole campaign in section 3. Summary and conclusions are shown in section 4.



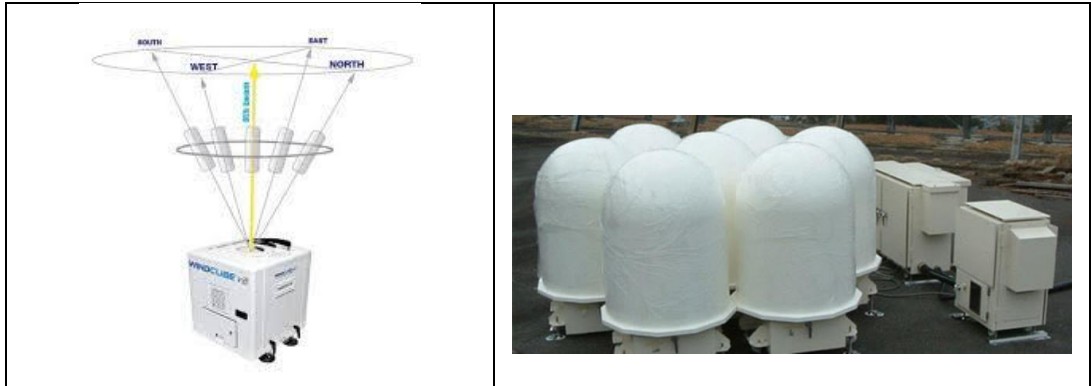

**Figure 1: (Left) The WindCube v2 Infrared Doppler Lidar, manufactured by Leosphere and provided by EKO Instruments Co., Ltd. (Japan) along with the 5 beam directions. (Right) The WPR LQ-7 antenna array.**


## 2. Computation and comparison methods

### 2.1 TS method

The accuracy of $\varepsilon$ and velocity variance estimates from the time series (TS) method is the subject of many sophisticated derivations (e.g. *Banakh et al. 2021* and references therein). They are particularly necessary for turbulence in the sheared

stratified atmospheric boundary layer and in presence of gravity waves (e.g. *Banakh et al. 2019*). In the present work, we use the following representation of the one-dimensional turbulence energy frequency spectrum of vertical velocity (e.g., *Banakh et al., 1999*):

$$S_w(f) \approx C_K \left(1 + \frac{1}{3} sin^2\alpha\right) \varepsilon^{2/3} \left(\frac{\overline{U}}{2\pi}\right)^{2/3} f^{-5/3} \tag{1}$$

where $C_K = 0.52$ is the Kolmogorov constant, $\overline{U}$ is the mean wind speed and $\alpha$ is the angle between the beam axis and the mean wind direction. For a horizontal wind and a vertical beam, the measurements are transverse ($\alpha = 90°$) so that $\left(1 + \frac{1}{3} sin^2\alpha\right) C_K = 4/3 C_K$. By default, we will apply this coefficient, even if it is not self-evident in the CBL because the horizontal wind speed $U$ can be of the order to or even smaller than the mean vertical velocities $W$, so that the measurements would not be transverse but longitudinal, for the asymptotic case $U \ll W$. In addition, in the case where the rms value $V_{rms}$ of

the turbulent wind fluctuations are not small compared to the mean wind velocity $\overline{U}$, the Taylor hypothesis would be violated. *Wilczek et al. (2014)* showed that the consequence is a dependence of $C_K$ with $\xi = V_{rms}/\overline{U}$ (their figure 1) and an underestimation of $C_K$ that can be of the order of 50 % for $\xi = 1$. In practice, we checked that this correction remains small for our observations. However, it should be kept in mind that the constant can deviate significantly from $4/3 C_K$ in extreme cases, i.e. when the mean wind is weak and not horizontal and turbulent velocity fluctuations are strong.

Strictly speaking, expression (1) is valid for point measurement spectra. *Banakh et al. (1999)* proposed to include the effects of the measurement volume by multiplying (1) by the transfer function $H(f)$ of a low-pass filter: $S_D(f) = S_w(f)H(f)$ (their expression 37 page 1051). For $(4\Delta z|sin\alpha|/\overline{U})f \gg 1$, where $\Delta z$ is the range resolution, the weighted spectrum is expected to vary as $f^{-8/3}$, not $f^{-5/3}$. In many cases, the condition for observing a -8/3 slope or a slope steeper than -5/3 was potentially met in our lidar data, especially when $\overline{U}$ is weak. However, we did not observe such a characteristic, but rather a very clear a

-5/3 slope when the horizontal wind is minimum. We will therefore restrict ourselves to using expression (1), keeping in mind that it is an approximation, widely accepted for other radar and lidar applications (e.g. *Hocking et al. 2016*).





W time series of 512 points for DL (64 points for LQ7), corresponding to ~34 min (63 min), are first detrended by removing a linear fit of the time series. They are then weighted by a (variance preserved) Hanning window. Because of the loss of energy at the edges of the time series resulting from the weighting function, a time oversampling by a factor of 2 is applied. The

frequency spectra $S(f)$ are then calculated using the fast Fourier transform (FFT) method. The Nyquist frequency is $0.125 \ s^{-1}$ for DL and $0.0085 \ s^{-1}$ for LQ7. The spectral levels in the $[0.01\text{-}0.08] \ s^{-1}$ frequency band for DL and $[0.002\text{-}0.007] \ s^{-1}$ frequency band for LQ7 are then obtained. $\varepsilon_{DL}$ and $\varepsilon_{LQ7}^{TS}$ are finally deduced by identification with the theoretical expression (1). This differs from *O'connor et al. (2010)* who calculated the total variance assuming that all the resolved scales lie within the inertial subrange. Here, we only need to assume that the selected and limited frequency band is consistent with the

Kolmogorov law. The consistency with a -5/3 slope has been tested by calculating the spectral slopes in the selected bands. They are obtained by (a) dividing the spectral band into two equal sub-bands (in log scale), (b) calculating the variance for each sub-band and (c) determining the slope of the straight line passing through these two points. The procedure described above was applied to frequency spectra and fixed spectral bands. We also applied it to wavenumber spectra $S(k) = 2\pi/\overline{U}S(f)$ and fixed wavenumber bands. The results were found to be very similar and are not shown. However, because the spectra are

not obtained with the same mean horizontal wind speed due to the altitude offset, we compared the wavenumber spectra $S(k)$ obtained from DL and LQ7.

## 2.2 DS method

The Doppler Spectral (DS) width method applied to LQ7 was described by *Luce et al. (2023a, b)*. It is based on the results presented by *Luce et al. (2018)* with the MU radar. We showed that a simple formulation:

$\varepsilon_{LQ7}^{DS} = \sigma^3/L$ (2)

where $L = 70$ m and $\sigma^2$ is the variance of Doppler spectral peaks caused by turbulence. Expression (2) provides the best statistical results of comparisons with $\varepsilon$ directly estimated from in situ measurements with Pitot sensors aboard DataHawk Unmanned Aerial Vehicles (UAVs) (*Lawrence et al., 2013, Kantha et al., 2017*). This apparent scale can be interpreted as the vertical integral length scale $l_w$ of vertical velocities W if $\sigma^2$ is the variance of W (*Albrecht et al., 2015*). It was later found

that expression (2) is a first approximation to a more general expression that depends on the depth of the turbulent layer: (*Luce et al., 2023b*):

$\varepsilon_{LQ7}^{DS} = \sigma^3/(0.1D)$ (3)

Where $D$ is the depth of the shear-generated turbulent layer. Note that Albrecht et al. (2015) found a similar result on average (i.e. $l_w \approx 0.1 \ D$) in the entrainment zone of a convective boundary layer topped by stratocumulus clouds. Lenschow and

Stankov (1986) found a dependence of $l_w/D$ with altitude in the convective boundary layer but 0.1 seems to be a typical value (their figure 2).

We will first test expression (2) for two reasons: It is very simple and does not require a priori information on the depth of the CBL so that it can be applied in real time if acceptable. Expression (3) is a hypothesis and requires a posteriori information on the CBL depth. We will consider this model in a second stage.

## 140 2.3 Comparison methods

As explained above, the DL and LQ7 comparisons were not made at the same height range, although the instrument parameters were set to allow measurements to be made up to 390 m for DL and from 300 m (height of the centre of the first radar gate) for LQ7. To assess the quality of the DL and LQ7 data, we tested the height continuity of daily averaged horizontal wind speeds (not shown). After removing spurious DL data due to low carrier to noise ratios and rain contaminations, we found a

negative bias (systematic for all days) for heights above 300 m for DL and for the first sampled height of LQ7 (i.e. 300 m). For these reasons, we discarded the corresponding data for DL and LQ7 and there is no longer any altitude overlap between




the two instruments. Our objective being to compare spectra of W and dissipation rates, which are expected to be rather uniform on average in the mixed layer of the CBL, we show results of observations between 07:00 LT and 17:00 LT enclosing the whole evolution of the CBL during the day. Outside this time window, the two instruments do not sample a priori the same

atmospheric dynamics due to the absence of height overlaps. In addition, LQ7 is highly sensitive to biological targets (birds, bats or insects) from the sunset to the sunrise. The raw data are strongly corrupted so that it is not possible to produce reliable estimates of atmospheric parameters at a time resolution of 59 s. Algorithms are used to discard these outliers in the 10-min averaged processed data.

To improve the precision of the estimated quantities (velocities, spectra, $\varepsilon$), we average these quantities in heights between

100 and 300 m for DL (i.e. 11 consecutive gates) and between 400 and 500 m (i.e. 2 consecutive gates) for LQ7 (see e.g. the black rectangular contour in Figure 3a), assuming that they are weakly variable in these height ranges during convection. This assumption has been verified in practice and examples will be shown.

## 3. Results

### 3.1 Meteorological background conditions

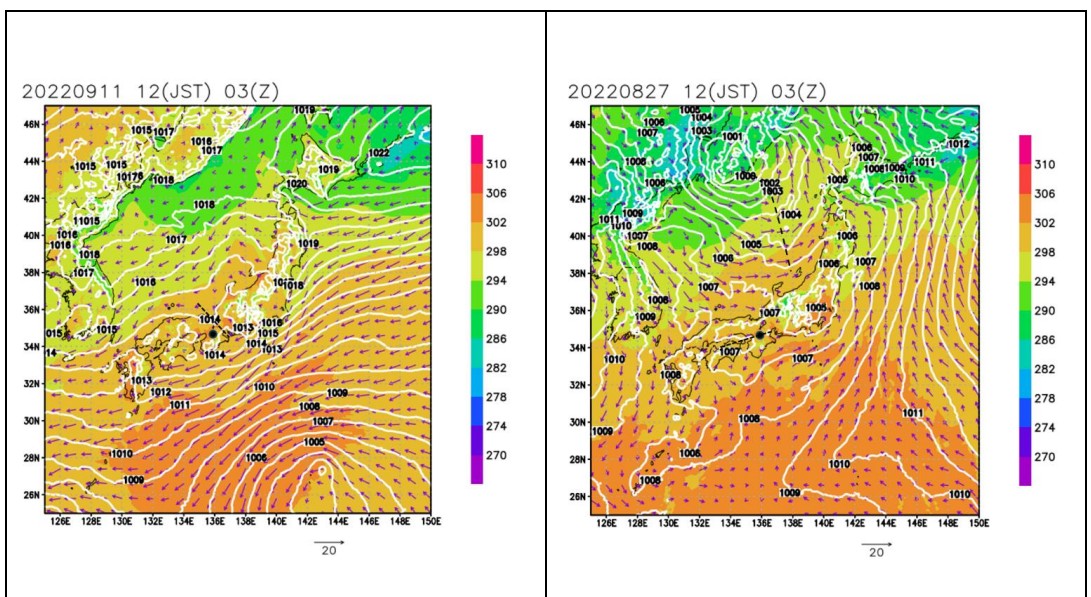

**Figure 2. Sea level pressure isobars, horizontal winds (vectors) and temperature (color levels) around Japan on 11 September 2022 (left) and on 27 August (right) at 12:00 LT. The black dot shows the Shigaraki MU Observatory location (GRADS/COLA).**

Figure 2 shows the contour plots of sea level pressure, surface horizontal winds (vectors), and temperature (color levels) around Japan on 11 September, 2022 (left) and 27 August (right) at 12:00 LT obtained from the Japan Meteorological Agency's (JMA) Grid Point Values (GPV) generated from the Meso Scale Model (MSM). These two days will be analysed in detail in section

3.2. A shallow low-pressure trough extending from a depression (~1000 hPa) centered over the Vladivostok region to the west of Japan, including the region of the Shigaraki MU Observatory (34°5N, 136.0°E), can be seen on 27 August. On 11 September, at 12:00 LT, high atmospheric pressure conditions (~1014 hPa) and a shallow ridge extending from a small anticyclone around (24°N, 138°E) were observed. The high-pressure conditions did not change until 15 September at least (day of the field campaign end). Two representative cloud cover conditions from a fisheye camera at 10:30 LT are shown in Figure 3 (left





panel: 27 August and right panel: 11 September). On 11 September, only fair weather cumulus clouds were present and the conditions were conducive to the formation of a convective boundary layer. On 27 August, stratiform clouds with a base located at about 2 km altitude were observed until ~17:00LT interspersed with short sunny spells. The low incoming solar radiation prevented the development of the CBL but on some occasions, lower level cumulus clouds (such as those in lower left of the photo) developed and dissipated.

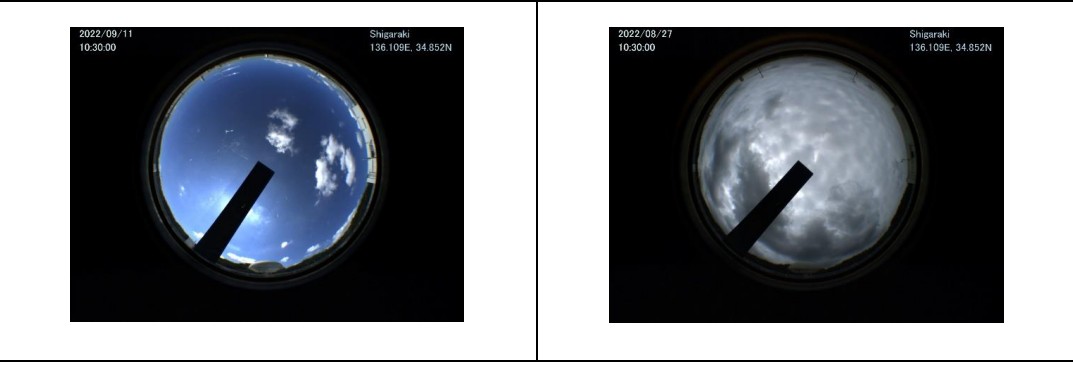

**Figure 3. Two examples of fisheye camera photos on 11 September and 27 August, 2022 at 10:30 LT at Shigaraki MU Observatory.**

Figure 4 shows time-height cross-sections of backscatter data measurements (in log10 scale) from a Vaisala CL31 ceilometer and hourly horizontal winds measured by DL averaged over 40-300 m height range, for 11 September 2022 and 27 August 2022. The ceilometer shows the vertical distribution of aerosols and cloud base. Approximately, the red levels indicate high concentrations of aerosols and the brown levels indicate clouds and (light) precipitation. The cloud base is highlighted by black

dots. The 910-nm ceilometer laser does not penetrate clouds.

On 11 September (upper panels), the evolution of the aerosol distribution during the day followed the standard evolution of CBLs with fair weather cumulus clouds developing above the mixed layer. The winds near the surface were dominantly southward during CBL development. The height of the cloud base is consistent with the height of the lifting condensation level (LCL) (blue curve) calculated from surface meteorological data collected at the observatory using Emanuel's (1994, pp 129-

130) derivations. LCL exceeded the height of 1.0 km in the afternoon. Also superimposed is a radar-derived CBL top height (thick black curve) at a vertical resolution of 100 m. This height is independently defined according to the height at which the LQ7 echo power begins to decrease, as it should correspond to the top of the turbulent entrainment zone of the CBL (e.g., *Angevine et al., 1994, Kumar and Jain, 2006*). The figure shows that there is a very close correspondence between the daily variations of the radar-derived CBL top height, LCL and the cumulus cloud base (when there is). The radar-derived CBL top

height is generally slightly higher than LCL and cloud base altitude except during the decaying stage of the CBL after ~14:00 LT where the CBL top height is slightly lower. This feature is sometimes more pronounced for other days (not shown). In the present study, the radar-derived CBL top height is used as a proxy of the CBL depth D. This estimate of D is not necessarily the one defined in expression (3) and may introduce a small uncertainty but it has the advantage of being obtained from the radar measurement alone.

On 27 August (lower panel of Fig. 4), a much shallower convective structure developed during daytime probably due to weaker solar heating. The stratiform cloud layer revealed by the fisheye camera picture (Fig. 3) was present from the height of ~2.0 km between ~07:00 LT and 15:00 LT during northward wind near the surface. The low-pressure synoptic conditions were likely favourable to the formation of convective plumes because of the systematic upward motions prevalent in the low-pressure systems. The lower cumulus clouds in the same image had a base of about 700 m between 10:00 LT and 11:00 LT,





relatively consistent with the LCL estimate. Another cumulus cloud development occurred after 15:00 LT. Associated with easterly winds, precipitation started after ~20:00 LT. Consistent with the ceilometer data, the LQ7 echo power measurements (not shown) do not show a CBL top evolution similar to 11 September. A CBL top was perhaps detected between 13:30 LT and 15:30 LT due to a late development caused by increasing incoming solar radiation from early afternoon sunshine spells. If we refer to LCL and to the altitude of aerosol concentration decrease after ~10:00 LT (lower panel of Fig. 4), LQ7 sampled

the upper part of the convective cells at the altitudes of 400-500 m selected for LQ7 analysis. It is therefore expected that the conditions are a priori not optimal for good agreements between $\varepsilon_{DL}$, $\varepsilon_{LQ7}^{TS}$ and $\varepsilon_{LQ7}^{DS}$.

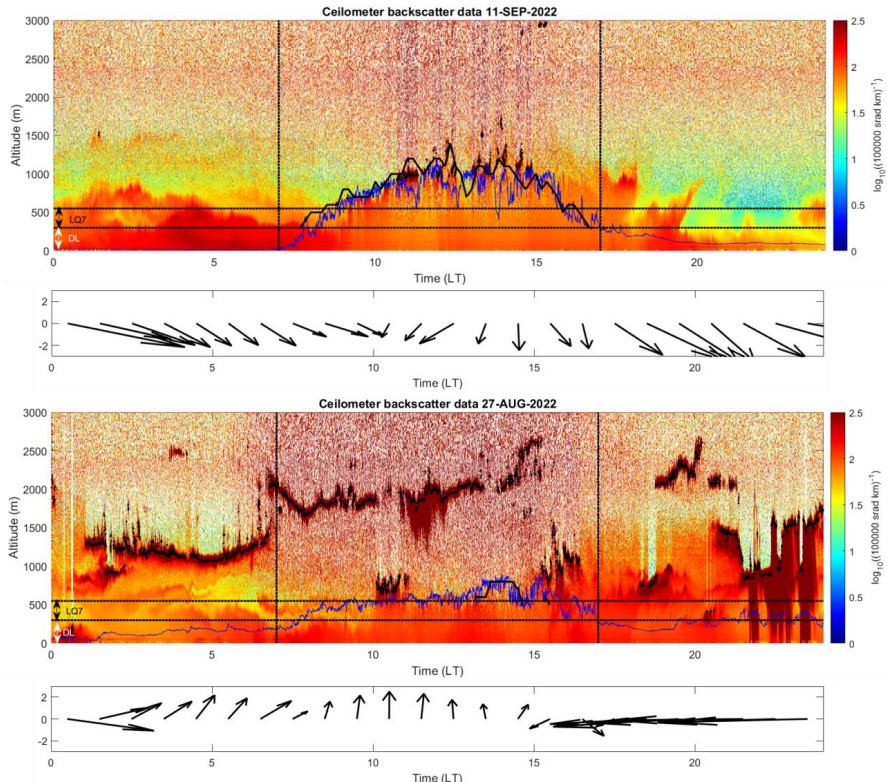

**Figure 4.** Time-height cross-sections of Vaisala ceilometer CL31 (910 nm) backscatter data (in $log_{10}$ scale) at a time and range resolution of 15 s and 20 m, respectively, from the ground to the height of 3000 m for 11 September, 2022 (Fig. 4a) and 27 August,

2022 (Fig. 4b). The black arrows indicate the hourly winds measured by the DL in the 40-300 m height range. The blue curve shows the height of the lifting condensation level (LCL) calculated from the surface meteorological data from the Automatic Weather Station (AWS). The thick black curve shows a proxy of the CBL top estimated from the identification of the maximum of the LQ7 echo power. The double arrows and horizontal lines indicate the range used to calculate the TKE dissipation rates from LQ7 and DL.

**3.2. Vertical velocity comparisons**

The upper panels of Figs 5a and 5b show time-height cross-sections of vertical velocities measured by DL in the [40-300] m height range above the ground level (AGL) and by LQ7 in the [400-1500] m height range from 07:00 LT to 17:00 LT for 11 September and 27 August, respectively. For this figure, the time series were smoothed with a 3-min rectangular window for both instruments to reduce the random noise fluctuations and the effect of the horizontal distance between the two instruments

(~80 m). The dashed curve in Fig. 5a is the proxy for the CBL top height shown in Fig. 4a.





On 11 September, the vertical velocities measured by DL and LQ7 show a remarkable height continuity. With a Pearson correlation coefficient of 0.57, the time series of $W$ from DL and LQ7 averaged over height (bottom panel of Fig. 4a) are very similar. Between 07:00 LT and 17:00 LT, the mean values of $W$ are 0.07 $ms^{-1}$ and -0.01 $ms^{-1}$ for DL and LQ7, respectively. The rms values, 0.57 $ms^{-1}$ and 0.65 $ms^{-1}$ for DL and LQ7, respectively, are very close. However, there are some significant

differences, for example, between 09:00 and 10:00 LT. Unfortunately, they cannot be interpreted due to the height offset. As cross-validation of the vertical velocity measurements is not possible, it is not investigated further in this paper and we assume that the differences are real (i.e. not instrumental). The $W$ fluctuations are significantly enhanced below the CBL (in particular between 10:00 LT and 14:00 LT, up to 2.2 $ms^{-1}$ and down to -1.7 $ms^{-1}$).

On 27 August, the $W$ fluctuations were weaker than those observed on 11 September. Between 07:00 LT and 17:00 LT, the

mean and rms values of $W$ are [-0.05, 0.30] $ms^{-1}$ and [-0.09, 0.26] $ms^{-1}$ for DL and LQ7, respectively. The correlation coefficient between the time series of $W$ from DL and LQ7 is 0.50, slightly less than for 11 September. Between 10:00 LT and 11:00 LT, when convection lead to cumulus cloud formation, the vertical velocities measured by DL and LQ7 are dominantly positive up to LCL (~700 m). Between 15:00 LT and 16:00 LT, when cumulus clouds also formed, stronger updrafts and downdrafts exceeding 1 $ms^{-1}$ are observed at least up to 1500 m, consistent with the very irregular pattern of the cloud base

in the ceilometer data (Fig. 4). The largest difference between the rms values of $W$ from DL and LQ7 is observed between these two events, i.e., between 11:00 LT and 15:00 LT: 0.34 and 0.18 $ms^{-1}$, respectively.

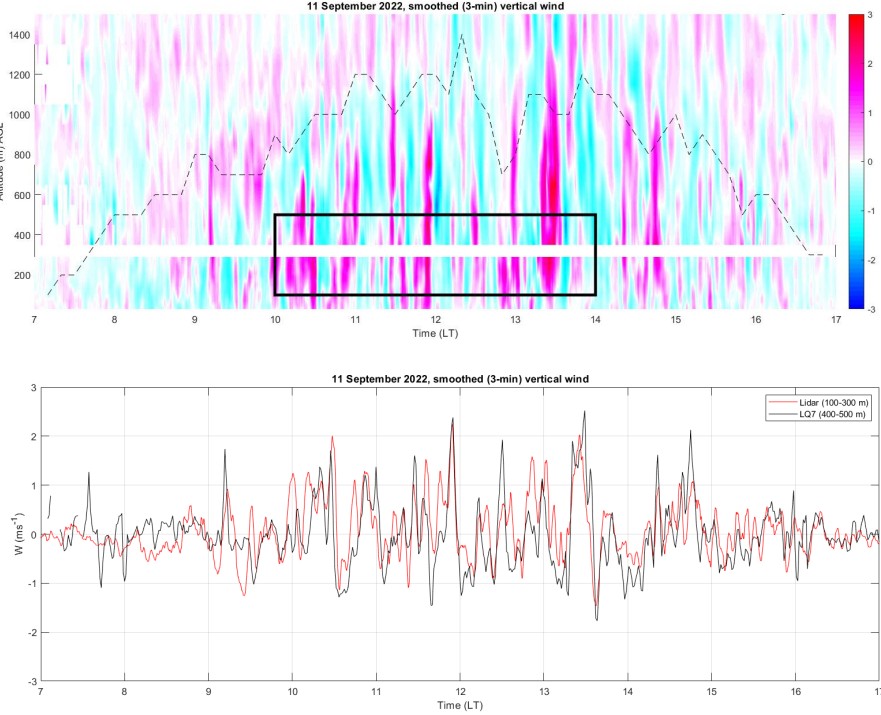

**Figure 5a: (top) Time-height cross-section of smoothed (3-min) values of vertical winds from LQ7 (400~) and DL (20-300 m) on 11 September, 2022 and from 07:00 LT to 17:00 LT. The proxy for the CBL top height is given by the dashed line. The black rectangle shows the time-height domain used to estimate W spectra shown in Figure 6. (Bottom) The corresponding time series of W for DL (red) averaged in the [100-300] m height range and for LQ7 (black) averaged over two consecutive sampled heights (400 and 500 m).**


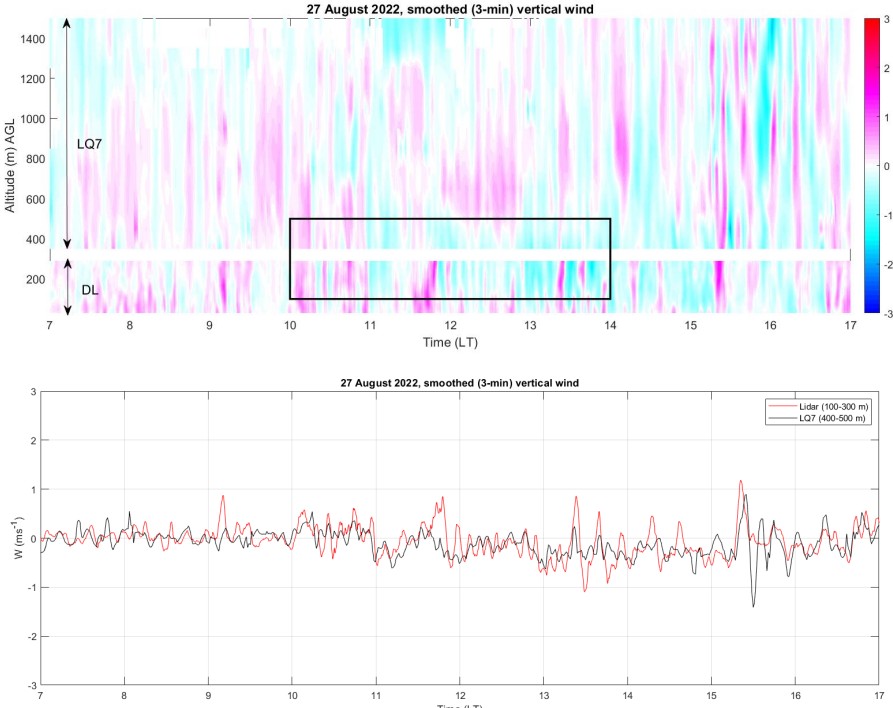

**Figure 5b: Same as Fig. 5a on 27 August, 2022.**

### 3.3 $\varepsilon_{DL}$ and $\varepsilon_{LQ7}^{TS}$ comparisons

Figure 6a shows the wavenumber spectra of $W$(DL) (blue) and $W$(LQ7) (black) for the data collected on 11 September between 10:00 LT and 14:00 LT, i.e., when the CBL was the deepest, using the procedures described in section 2. The red dashed line shows the theoretical inertial spectrum ($-5/3$ slope) and the blue dotted curve, the best fit of the DL spectrum with the theoretical *Kristensen et al. (1989)* 1-D line spectrum using $\mu = 1$ (see *Lothon et al (2009)* for a more detailed description and the definition of the parameter $\mu$). A good fit indicates that the calculated spectrum is consistent with the turbulence spectrum expected for a CBL. Even if the calculated slopes are flatter than $-5/3$, the two spectra are not inconsistent with an inertial subrange as suggested by the red dashed line. The LQ7 spectrum agrees well with the high wavenumber part of the DL spectrum in slope, shape and level suggesting that the same turbulent regime was detected by both instruments. In this case, we therefore expect consistent values of $\varepsilon_{DL}$ and $\varepsilon_{LQ7}^{TS}$.

Figure 6b shows the corresponding wavenumber spectra for data collected on 27 August. Note that the two wavenumber ranges of the spectra overlap more than those shown in Fig. 6a because the wind speed was ~2.5 $ms^{-1}$ in both altitude ranges selected for DL and LQ7. It can be seen that: (1) the two spectral levels are significantly lower than those for 11 September (Fig. 6a) and the deviation from $-5/3$ for the DL spectrum occurs at higher wavenumbers ($k > 2\ 10^{-2}\ rad\ m^{-1}$). (2) The LQ7 spectrum shows a level 3 to 4 times lower than the DL spectrum, a lower level consistent with the weaker fluctuations and smaller rms values of $W$ shown in Fig. 5b. Incidentally, the LQ7 spectrum exhibits a peak at $k_m \sim 0.07\ rad\ m^{-1}$, a $-5/3$ slope at higher wavenumbers and a flat spectrum at lower wavenumbers down to $2\ 10^{-3}\ rad\ m^{-1}$. This resembles an inertia gravity wave spectrum for $k < k_m$ for quiet conditions, as a flat spectrum (i.e., 0 slope) is expected from gravity wave theory (e.g.,



*VanZandt, 1982*). However, this interpretation does not seem consistent with the context of convection described in section 3.1 and this hypothesis will not be explored further.

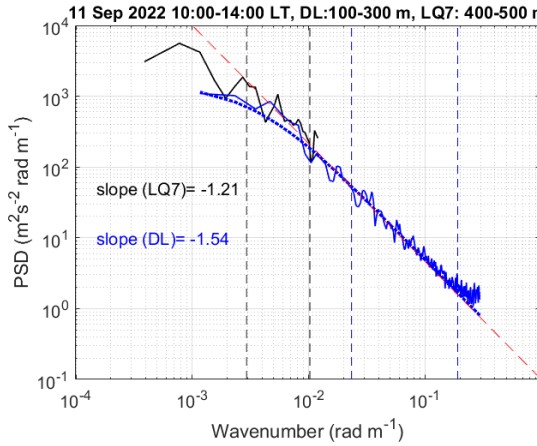

**Figure 6a. Mean wavenumber spectra of W obtained on 11 September, 2022 for 10:00-14:00 LT from DL 4-s time series (blue) between heights of 100 m and 300 m and LQ7 59-s time series between heights of 400 m and 500 m. The frequency to wavenumber conversion was made using the mean horizontal wind speed: 2.6 $ms^{-1}$ between 100 and 300 m and 4.3 $ms^{-1}$ between 400 and 500 m. The red dashed line shows the -5/3 slope for reference. The calculated slopes are for the wavenumber ranges indicated by the vertical dashed lines.**

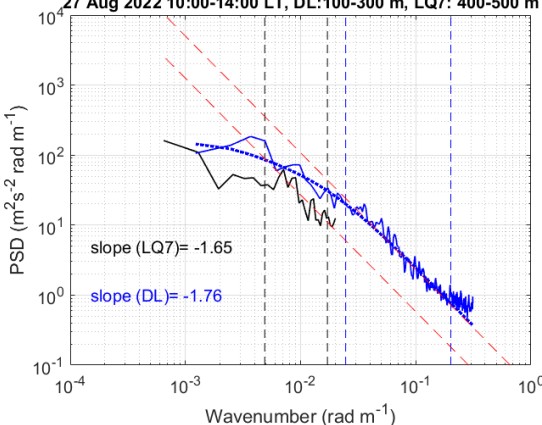

**Figure 6b. Same as Fig. 6a for 27 August, 2022. The frequency to wavenumber conversion was made using the mean horizontal wind speed: 2.5 $ms^{-1}$ between 100 and 300 m and 2.6 $ms^{-1}$ between 400 and 500 m. The red dashed lines show the -5/3 slope for reference. The calculated slopes are for the wavenumber ranges indicated by the black and blue vertical dashed lines.**

Figure 7 shows the time series of $log_{10}(\varepsilon_{DL})$ (blue) averaged in the [100-300] m height range and $log_{10}(\varepsilon_{LQ7}^{TS})$ (black) averaged in the [400-500] m height range at a time resolution of 1 hour every 30 min for 11 September (top panel) and 27 August (bottom panel). As the time resolution of $\varepsilon_{DL}$ is 30 min every 15 min, the two values for each hour were averaged. $\varepsilon_{LQ7}^{TS}$ before 07:00 LT and after 17:00 LT is not shown because corrupted by biological targets or rain echoes. On 27 August, $\varepsilon_{DL}$ was also corrupted by precipitation after ~20:00 LT.

On 11 September, the increased dissipation rate (typically $10^{-3}\ m^2s^{-3}$) associated with turbulence in the CBL is clearly visible, and $\varepsilon_{DL}$ and $\varepsilon_{LQ7}^{TS}$ time series show similar trends and levels. The most significant differences between the two estimates are observed before 08:30 LT and after 15:30 LT, i.e. during the development and decay phases of the CBL, and when the





CBL top height was less than 600 m. On 27 August, dissipation rates are significantly weaker than on 11 September (typically

~3 $10^{-5}$ to 3 $10^{-4}$ $m^2 s^{-3}$). In addition, as expected from the spectral levels (Fig. 6b), $\varepsilon_{LQ7}^{TS}$ is weaker than $\varepsilon_{DL}$ by a factor

consistent with $3^{3/2} \sim 5$ to $4^{3/2} = 8$ in average. This result may indicate that either the conditions are not met for reliable

estimates from the TS method applied to LQ7 data (assuming that $\varepsilon_{DL}$ is a reference), or that dissipation rates are indeed

significantly lower above 400 m altitude, or both. The largest discrepancies are obviously expected when the convection (or

CBL) top height is too low (compared with the altitude range used for LQ7). In order to check this assertion, we made a

295 statistical comparison between $log_{10}\left(\varepsilon_{LQ7}^{TS}/\varepsilon_{DL}\right)$ (i.e. the differences between the black and blue curves in Fig. 7) and the CBL

depths. Figure 8 shows the scatter plot of $log_{10}\left(\varepsilon_{LQ7}^{TS}/\varepsilon_{DL}\right)$ vs CBL depth (m) for all the available data between 21 August and

15 September after removing all data corrupted by precipitation echoes. The red curve shows $< log_{10}\left(\varepsilon_{LQ7}^{TS}/\varepsilon_{DL}\right) >$ obtained

by averaging $log_{10}\left(\varepsilon_{LQ7}^{TS}/\varepsilon_{DL}\right)$ over segments of 200 m. $\varepsilon_{LQ7}^{TS}$ is statistically weaker than $\varepsilon_{DL}$ for all CBL depths. The dispersion

is maximum and $< log_{10}\left(\varepsilon_{LQ7}^{TS}/\varepsilon_{DL}\right) >$ is minimum (most negative) when the CBL depth is minimum: It is a generalization

of the results obtained for 27 August. In contrast, the dispersion is minimum and $< log_{10}\left(\varepsilon_{LQ7}^{TS}/\varepsilon_{DL}\right) >$ is less than a factor of

2 when $D > 1000\ m$. Therefore, when DL and LQ7 sample CBLs deeper than 1000 m, $\varepsilon_{LQ7}^{TS}$ is statistically larger than $\varepsilon_{DL}$ by

no more than a factor of 2 in their respective height ranges (100-300 m and 400-500 m).

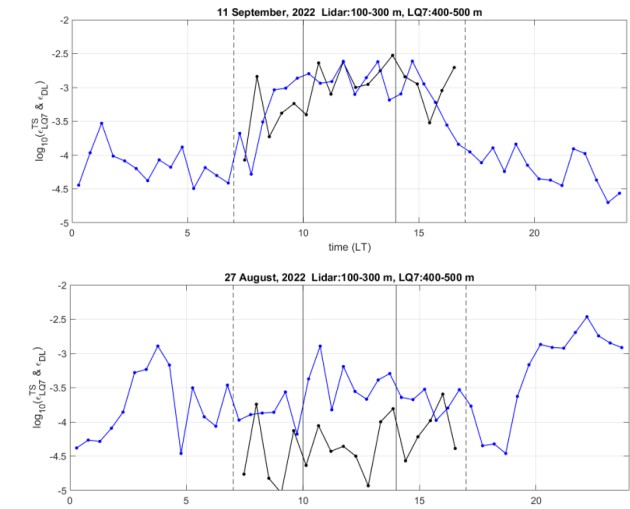

**Figure 7: Time series of $log_{10}(\varepsilon_{LQ7}^{TS})$ (black) in the [400-500] m height range and $log_{10}(\varepsilon_{DL})$ (blue) in the [100-300] m height range**

**on 11 September, 2022 (top) and 27 August, 2022 (bottom) at a time resolution of 1 hour sampled every 30 min. The vertical solid**

**black lines show the [10:00-14:00] LT time range used to compute the mean spectra shown in Fig. 6. The vertical dashed black lines**

**show the [07:00-17:00] LT time range shown in Fig. 5. $\varepsilon_{LQ7}^{TS}$ before 07:00 LT and after 17:00 LT is not shown because the raw LQ7**

**data are strongly corrupted by biological targets (birds and/or bats).**



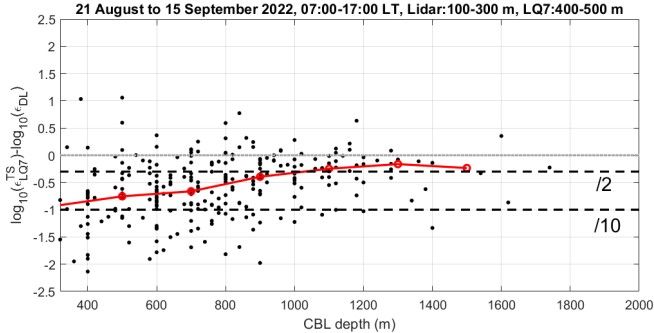

**Figure 8: Scatter plot of 1-hour averaged $log_{10}(\varepsilon_{LQ7}^{TS}/\varepsilon_{DL})$ vs CBL depth (m) for 23 days during the period 21 August to 15 September, 2022 excluding periods of precipitations. The red curve shows the trend obtained by averaging $log_{10}(\varepsilon_{LQ7}^{TS}/\varepsilon_{DL})$ over segments of 200 m.**

### 3.4 $\varepsilon_{DL}$ and $\varepsilon_{LQ7}^{DS}$ comparisons.

Since the comparison of $\varepsilon_{DL}$ and $\varepsilon_{LQ7}^{DS}$ is the main objective of the present work, initiated by the results of Luce et al. (2018, 2023a,b), we present the results for more CBL case studies. The left panel of Fig. 9a shows time-height cross-sections of $log_{10}(\varepsilon_{DL})$ and $log_{10}(\varepsilon_{LQ7}^{DS})$ using expression (2) with $L = 70\ m$ for four consecutive days from 11 September to 14 September when the deepest CBLs were observed. The grey translucent rectangles partially hide the values of $\varepsilon_{LQ7}^{DS}$ when they are corrupted by biological targets and rain echoes. The LQ7-derived CBL top heights are represented by the thick black curves. The right panel of Fig. 9a shows the corresponding profiles averaged between 07:00 LT and 17:00 LT. The horizontal bars indicate the standard deviations for each height sampled by the two instruments. The most striking feature is the high consistency of the $\varepsilon_{LQ7}^{DS}$ and $\varepsilon_{DL}$ patterns for each day. Between 300 m and 400 m, there are virtually no discontinuities in height between the two mean profiles. The largest discrepancy is observed on 12 September but it does not exceed a factor of 2 ($log_{10}(\varepsilon_{DL})(300\ m) = -2.7$ and $log_{10}(\varepsilon_{LQ7}^{DS})(400\ m) = -3.0$). Moreover, this can largely be explained by the very low availability of DL data (~10%) in the highest DL gates, so that the effective time average for $\varepsilon_{DL}$, over 1 hour rather than 10 hours, may not be representative of the whole period. Therefore, our results suggest that the DS method (with $L = 70\ m$) applied to a UHF radar can provide estimates of dissipation rates in the CBL that are very consistent with lidar estimates with the TS method. Given (1) the different methods of analysis, (2) the different nature of the data and instruments, (3) the arbitrary nature of the choice of $L$, and (4) the difference between the height ranges, such a result was hardly expected.

Even more surprising are the results for 27 August 2022 (Fig. 9b). The time-height cross-section of $log_{10}(\varepsilon_{DL})$ and $log_{10}(\varepsilon_{LQ7}^{DS})$ also shows a good continuity in height despite a weaker and more fluctuating pattern. In particular, increased dissipation rates around 10:30 LT and 15:30 LT, during which cumulus clouds formed, and 12:00 LT, 13:00 LT are detected by both instruments without substantial anomalies in level. The mean profiles of $log_{10}(\varepsilon_{DL})$ and $log_{10}(\varepsilon_{LQ7}^{DS})$ show an almost linear decrease with height up to ~1000 m, except for $log_{10}(\varepsilon_{DL})$ in the [240-300] m height range and this feature may be due to the reduced data availability (the 50 % of $\varepsilon_{DL}$ values that are missing occur when $\varepsilon_{LQ7}^{DS}$ is minimum). Although the quantitative agreement may be partly coincidental, it is unlikely that the detection of the successive maxima of dissipation rates by the two instruments in their respective height ranges is purely coincidental. In addition, similar agreements are also found for other similar days (not shown). Therefore, the DS method applied to LQ7 data gives much better agreement with the TS method applied to DL data than the TS method applied to LQ7 data for shallow convective layers.





Incidentally, the increased dissipation rates above cloud base up to ~1200 m around 10:30 LT and, even more so, up to ~2000

345 m around 15:30 LT ($log_{10}(\varepsilon_{LQ7}^{DS})$ greater than -2) are not the result of outliers and are associated with strongly enhanced spectral widths (not shown) alternating with updrafts and downdrafts (see Fig. 5b).

By averaging over 10 hours (07:00 LT -17:00 LT), the dissipation rates in the CBL are mixed with lower values estimated above the CBL, so that the mean $\varepsilon_{LQ7}^{DS}$ profiles decrease smoothly with height in the range of the CBL. However, peaks of mean $\varepsilon_{LQ7}^{DS}$ values are also observed above the CBL. They are mainly distributed in layers (e.g. ~1600 m on 11 September, ~2500 m

on 12 September). They are not (at least directly) related to CBL dynamics and are probably due to other mechanisms, such as shear instabilities or other convective instabilities associated with clouds.

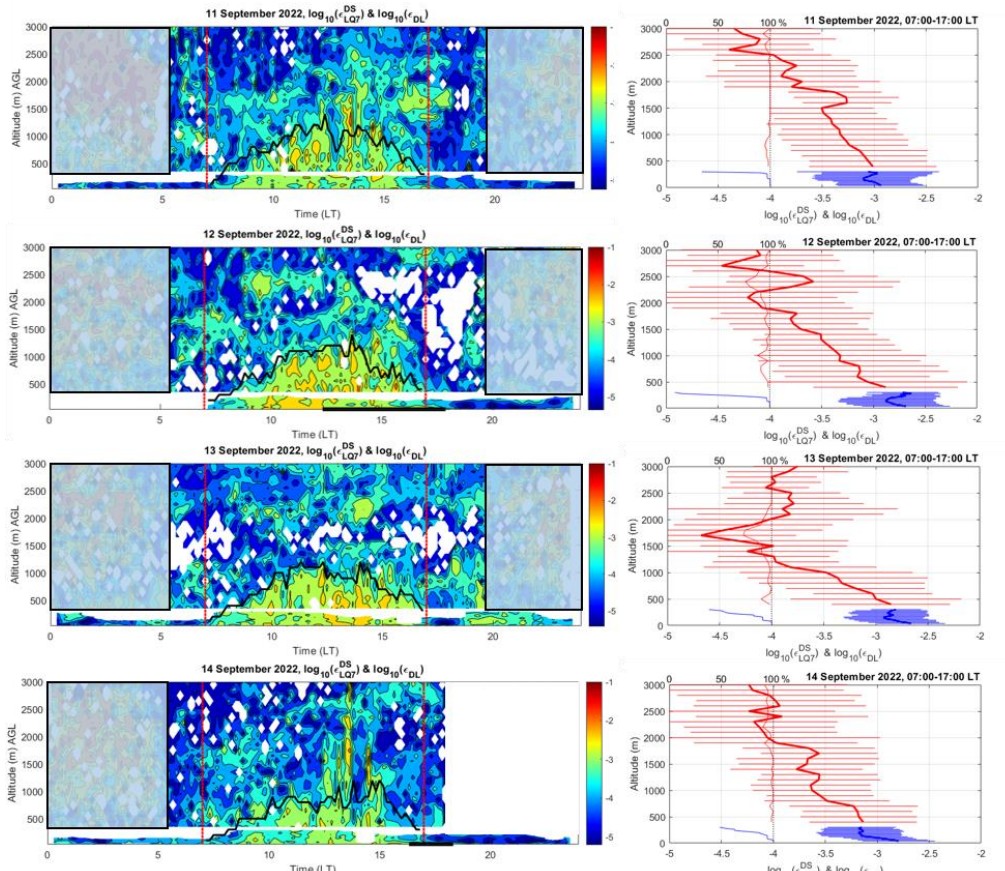

**Figure 9a: (Left)** Time-height cross-section of $log_{10}(\varepsilon_{LQ7}^{DS})$ estimated from eq. 2 and $log_{10}(\varepsilon_{DL})$ for 4 successive days from 11

September to 14 September 2022 up to the height of 3000 m. The translucent rectangles in $log_{10}(\varepsilon_{LQ7}^{DS})$ cover areas strongly affected by biological targets. The bold solid line is a proxy for the CBL top height. The two vertical red lines mark 07:00 and 17:00 LT. **(Right)** The corresponding profiles of $log_{10}(\varepsilon_{LQ7}^{DS})$ (red) and $log_{10}(\varepsilon_{DL})$ (blue) averaged over 10 hours between 07:00 and 17:00 LT. The horizontal lines show the standard deviation. The thin red and blue curves show the data availability between 0 and 100 % for each instrument.



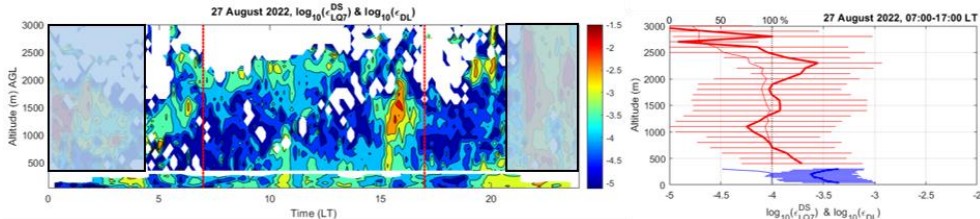

**Figure 9b: Same Fig. 9a for 27 August 2022.**

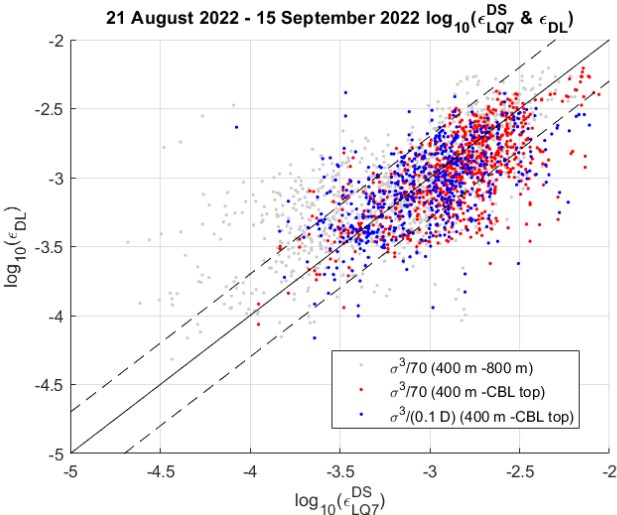

**Figure 10: Scatter plot of range (100-300 m) and hourly mean $log_{10}(\varepsilon_{DL})$ using the TS method vs hourly mean $log_{10}(\varepsilon_{LQ7}^{DS})$**
**averaged between 400 m and 800 m with $L = 70\ m$ (grey dots), between 400 m and the (variable) altitude of the CBL top (red dots)**
**and between 400 m and the (variable) altitude of the CBL top with $L = 0.1\ D$ (blue dots) in the [07:00-17:00] LT time range and for**
**23 days between 21 August and 15 September 2022, excluding rain periods.**

Figure 10 shows the scatter plot of hourly averaged $log_{10}(\varepsilon_{DL})$ and $log_{10}(\varepsilon_{LQ7}^{DS})$ for all data available from 27 August to 15
September (i.e., 23 days) between 07:00 LT and 17:00 LT, after excluding data from rain periods. As for the previous figures,
the $\varepsilon_{DL}$ values have been obtained after averaging in the [100-300] m height range. The $\varepsilon_{LQ7}^{DS}$ values have been averaged over
height in two ways to highlight the fact that the high correlation between the two estimates is not coincidental: (a) between
400 and 800 m (grey dots), ignoring the time variations in CBL top height and (b) between 400 and CBL top height (red dots).
When the average is limited to the CBL top height, the dispersion is significantly reduced, especially for $log_{10}(\varepsilon_{LQ7}^{DS}) < -3.5$.
This is due to the rejection of values outside the boundary layer when the height of the CBL is less than 800 m. The blue dots
show the results using (c): expression (3) with $L = 0.1\ D$, averaged between 400 m and the CBL top height. The scatter plot
is very similar to the scatter plot with (b). For both cases (b) and (c), ~80% of the estimates differ by less than a factor of 2,
confirming the conclusions made from the case studies (Fig. 9).
The fact that expression (2) with $L = 70$ m and expression (3) with $L = 0.1\ D$ lead to the same statistical agreement indicates
that they should not differ much. Indeed, as shown by Fig. 11, the distribution of CBL depth is between 300 and 1200 m, with
a mean value of 770 m. Therefore, the mean value of $L = 0.1\ D$ is 77 m, i.e. very close to the value used by default (70 m).
We can therefore interpret the value of 70 m for the CBL as a fraction of a typical boundary layer depth, i.e., one tenth according
to our "radar definition" of the CBL depth. Since expression (3) was established for turbulent layers generated by Kelvin-
Helmholtz (shear flow) instabilities (*Luce et al., 2023b*) supposed to be associated with small Richardson numbers ($Ri \ll 1$),

 

it can be concluded that this expression does not depend on the nature of the instabilities and is thus universal as long as we

can overlook the effect of the stable stratification, i.e. for $Ri \ll 1$ or negative).

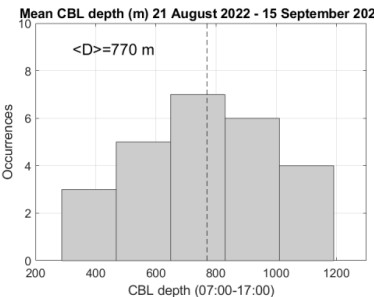

**Figure 11: Histogram of CBL depth between 07:00 and 17:00 LT for 23 days between 21 August and 15 September 2022.** $< D >=$

**770 $m$ is the mean value.**

Figure 12 shows the histograms of $log_{10}\left(\varepsilon_{LQ7}^{DS}/\varepsilon_{DL}\right)$ for (a), (b) and (c). The statistics for each case (mean and standard

deviation) are quite similar. However, the histogram for (a) is asymmetric due to the inclusion of low $\varepsilon_{LQ7}^{DS}$ values from above

the CBL top height. The histogram for (b) is also slightly asymmetric. The Chi-Square test for normality applied to the three

histograms indicate that only distribution (c) is log-normal. This property may confirm a dependence of $L$ with the depth of

the CBL because a log-normal law is expected when comparing two methods of estimation of the same parameter (or there is

an estimation bias).

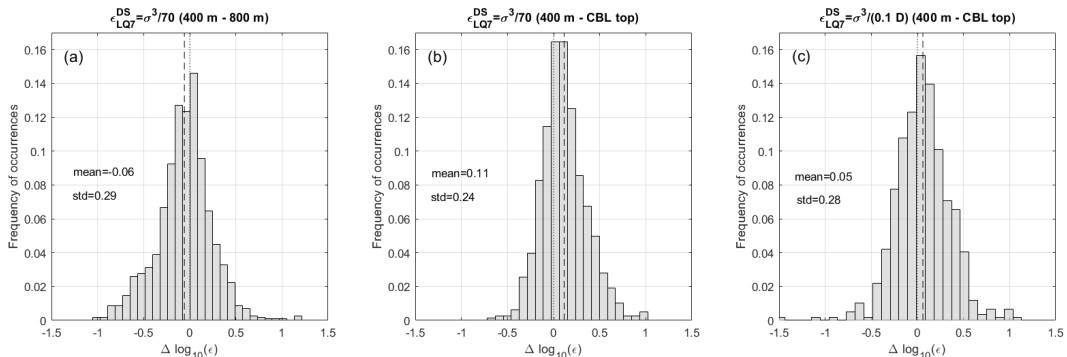

**Figure 12: Histogram of the difference $log_{10}\left(\varepsilon_{LQ7}^{DS}\right) - log_{10}(\varepsilon_{DL})$ for (a) $\varepsilon_{LQ7}^{DS} = \sigma^3/70$ averaged between 400 and 800 m, (b)**

**$\varepsilon_{LQ7}^{DS} = \sigma^3/70$ averaged between 400 and the CBL top height, (c) $\varepsilon_{LQ7}^{DS} = \sigma^3/(0.1\ D)$ averaged between 400 and the CBL top height.**

The mean value $log_{10}\left(\varepsilon_{LQ7}^{DS}\right) - log_{10}(\varepsilon_{DL})$ for (c) is 0.05, ie the ratio is 1.12 in linear scale. The mean values of $\varepsilon_{LQ7}^{DS}$ and

$\varepsilon_{DL}$ are thus virtually identical, suggesting that the dissipation rates are nearly uniform in the whole column of the CBL (above

the surface layer). However, it is known from theory and observations that TKE dissipation rates slightly decrease with height.

Figure 13 shows an experimental evidence from measurements obtained by DataHawk UAVs at Shigaraki MU Observatory

(*Luce et al., 2020*) and from a Doppler Lidar in USA (*de Szoeke et al., 2021*). The two profiles have been normalized using

the standard normalization procedures for CBL dynamics (*Stull, 1988*). The reader may find more information in the

aforementioned references. Here, we focus on the fact that $\varepsilon$ should decrease by a factor of ~2 at least in the mixed layer. The

slightly lower levels of $\varepsilon_{LQ7}^{TS}$ with respect to $\varepsilon_{DL}$ (a factor of ~2) described in section 3.2 when the CBL is deep are very





consistent with this property. Therefore, we cannot exclude that the equivalence between $\varepsilon_{LQ7}^{DS}$ and $\varepsilon_{DL}$ may be fortuitous and may be hide a slight uncertainty in the relationship between $L$ and $D$. Therefore, from our analyses, we propose:

$$\varepsilon_{LQ7}^{DS} = \sigma^3/(\alpha D) \qquad (4)$$

with $0.1 < \alpha < 0.2$.

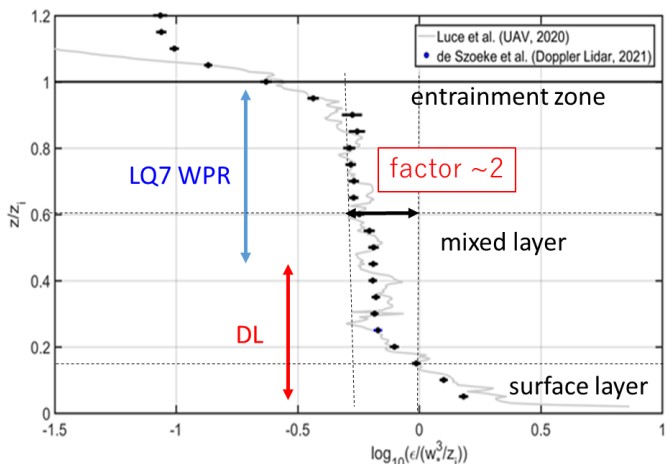

**Figure 13: Vertical mean profiles of normalized TKE dissipation rates measured by DataHawk UAVs (*Luce et al., 2020*) and by a Doppler lidar (*De Szoeke et al., 2021*). The range of observations from DL and LQ7 are indicative.**

## 4. Summary and conclusions

In the present work, we compared the estimates of TKE dissipation rates from a Doppler lidar (DL) and a UHF wind profiler (LQ7). Due to the lack of range overlap, comparisons make sense for convective boundary layers only, for which we expect some degree of homogeneity with height in the mixed layer. For DL, we used a method based on the calculation of 1-D frequency spectra, already tested in the literature. $\varepsilon_{DL}$ constitutes the reference for the comparisons with dissipation rates estimated from LQ7, even though it has not been validated by comparisons with independent in situ measurements (such as

those carried out by *Luce et al. (2018)* with the MU radar and instrumented UAVs, for example). However, the assumption that DL and the analysis method give reliable values is a posteriori accepted in view of the agreements obtained with LQ7. For LQ7, we used the method based on frequency spectra ($\varepsilon_{LQ7}^{TS}$), despite of the poor time resolution (59 s instead of 4 s for DL) and the more commonly used method based on the Doppler spectral width ($\varepsilon_{LQ7}^{DS}$). In general, the former has to be avoided in the free atmosphere because it is expected to be contaminated by gravity waves even for frequencies higher than N due to

Doppler shift effects. For the first time, we tested the method in CBLs during anticyclonic conditions and low-pressure cloudy conditions. We found that $\varepsilon_{LQ7}^{TS}$ give similar results on some occasions when the CBL is deep and persistent so that the LQ7 frequency spectra can be representative of turbulence in the inertial subrange. When the depth D exceeds 1000 m, hourly estimates of $\varepsilon_{LQ7}^{TS}$ in the [400-500] m height range are statistically consistent with hourly averaged $\varepsilon_{DL}$ in the [100-300] m height range and smaller by a factor of 2 or less. This small difference may be significant because dissipation rate is expected

to decrease slightly with height in the mixed layer. This result was useful for the interpretation of the results obtained with spectral width method. We first applied the model $\varepsilon_{LQ7}^{DS} = \sigma^3/L$ with a constant $L = 70\ m$, in accordance with the results found by *Luce et al. (2018, 2023a, b)* for shear generated turbulent layers. A good agreement with $\varepsilon_{DL}$ was found despite the



disconcerting simplicity of the model, even for shallow convective layers (depth <~600 m). An even better agreement was obtained with $\varepsilon_{LQ7}^{DS} = \sigma_t^3/(0.1\,D)$ where $D$ is the CBL depth. The relevance of the simple model with $L = 70\,m$ is due to the

fact that the numerical value is close to one tenth of the typical value of the CBL depth. A similar observation was made by Luce et al. (2023b) as $L \sim 70$ m is also one tenth of the typical thickness of turbulent layers generated by shear instabilities detectable by LQ7 in the troposphere. For much deeper CBLs (e.g., 2000 m or more), $\varepsilon_{LQ7}^{DS} = \sigma_t^3/(0.1\,D)$ should be more adapted than $\varepsilon_{LQ7}^{DS} = \sigma_t^3/70$. The quantitative agreement between $\varepsilon_{LQ7}^{DS}$ and $\varepsilon_{DL}$ should conceal a disagreement, since the two estimates are not obtained in the same range and the dissipation should decrease with height, as shown by the comparisons

between $\varepsilon_{LQ7}^{TS}$ and $\varepsilon_{DL}$. Therefore, $\varepsilon_{LQ7}^{DS} = \sigma_t^3/(0.1\,D)$ is uncertain and a coefficient between 0.1 and 0.2 should be considered instead. A clarification of this point should be obtained by comparing dissipation rates in the same altitude range with another Doppler lidar and another wind profiler. More decisive results may be obtained with a Doppler lidar providing spectral width as well.

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

*Author contributions*. HL and MY acquired and processed the data and wrote the paper.

*Competing interests*. The authors declare that they have no competing interests.

*Financial support*. This study was partially supported by JSPS KAKENHI grants 19H04238, 20K21844 and 22H03732.