# Peer review of "Turbulence kinetic energy dissipation rate estimated from a WindCube Doppler Lidar and the LQ7 1.3 GHz radar wind profiler in the convective boundary layer."

_Atmospheric Measurement Techniques, 2024_

## Author Comment (AC1)

Response to reviewer 2

This manuscript builds on previous papers (by Luce and various co-authors) that have developed a novel formulation for extracting turbulence kinetic energy dissipation rates from wind profiling radar observations. The earlier papers looked at shear generated turbulence within the free troposphere as observed by a lower-VHF radar. The present manuscript extends the analysis to turbulence occurring within the Convective Boundary Layer (CBL). It compares observations made by a UHF radar and a Doppler Lidar. Despite the fact that there is no altitude overlap between the two sets of measurements, the manuscript shows that the data are consistent when the CBL is well developed, but differ when the CBL is less well developed.

This is an interesting manuscript. The analysis and discussion are sufficiently thorough that I do not have substantial questions about the science. There are, however, a few points that would benefit from clarification. Most of my comments relate to the figures, since some aspects were not entirely clear when the manuscript was printed out at A4 size.

We sincerely appreciate the time and effort the reviewer took to evaluate our manuscript and for his positive assessment.

POINTS REQUIRING CLARIFICATION

1) On line 53, it is stated that LQ7 observations are made at a time resolution of 59 s and that the data are also processed to provide deliverables at a time resolution of 10 minutes. It would be helpful if the authors could clarify when they are using 59 s data (presumably for the time series analysis) and when they are using 10 minute data.

We have clarified this point in Sections 2.1 and 2.2, where the TS and DS methods are described, as these methods have not yet been introduced in the Introduction. Specifically, the TS method utilizes data collected at a time resolution of 59 s, while the DS method relies on the (10-min) processed data of spectral width and horizontal wind speed (the latter being used for beam-broadening correction). The 59-second time resolution data can also be used for the DS method; however, our intention was to show that the publicly available 10-minute data can be used as well, making it easier for interested readers to verify the method.

2) Do references, on line 90, to the mean wind (U bar) and the angle between the beam axis and mean wind direction (α) imply three-dimensional measures or just horizontal ones?

According to Banakh et al. (1999), the mean wind refers to the three-dimensional wind $\vec{U}$ $(u, v, w)$. Strickly speaking, α is the angle between the beam direction and the direction of the 3D wind.

3) What does the word "minimum" imply on line 105 in the sentence, "However, we did not observe such a characteristic, but rather a very clear -5/3 slope when the horizontal wind is minimum."? Does this imply slow wind speeds or actually to the minimum values?

The word "Minimum" is indeed ambiguous and we replaced it by "weak (~2-5 m/s)"

4) In section 2.2, it would be useful to add a paragraph that summarises the spectral width method, e.g. indicating that vertical beam observations corrected for the effects of beam

broadening have been used. At present, the reader must refer to the earlier papers for these details.

Following the suggestions of both reviewers, the methodology for applying the spectral width technique has been expanded.

5) For completeness, the following abbreviations should be given in full where they first appear in the main manuscript. I note that two of them are given in the abstract.

5a) "CBL" on line 46; it is given on line 47 (and on line 21 in the abstract)

5b) "UHF" on line 48

5c) "TS" on line 61 (given on line 17 in the abstract)

5d) "VHF" on line 62

5e) "rms" on line 94

The acronyms are now defined

FIGURES

6) The axis labels in Figure 2 are difficult to read since they are so small. Using a non bold font might help. Otherwise it would be useful to mark the location of the observation site with a marker.

Done

7) The tick labels on the time axis of Figures 4, 7, and 9 are quite far apart (5 hour intervals). It would be helpful for the reader if these could be shown at shorter intervals (c.f. Figure 5) since the manuscript refers to features that occur at specific times. It would also help if the tick marks could be shown pointing outwards from the plot area.

We have replotted with a 1-hour time interval.

8) For the right hand side of the bottom panel of Figure 4, the wind direction arrows overlap each other making it difficult to see if they represent easterly or westerly winds (I note that this is clarified in the main text).

The scales have been adjusted to shorten the arrows, and a scale reference has been added. Although the wind direction still overlaps after 16:00 LT in Fig. 4b, we hope this version is more readable.

9) For clarity, it would be helpful to state the approximate backscatter values corresponding to the "red" and "brown" colours in Figure 4, which are referred to on lines 178 and 179. I think what the authors refer to as "brown" is what I would describe as dark red.

The text has been revised to include quantitative levels.

10) It can be difficult to distinguish between the blue and black lines on the upper two panels of Figure 4 and in Figure 6a. Using higher contrast colours could help. It is also difficult to distinguish between the grey and blue dots in Figure 10. Using larger dots might help.

The blue line has been replaced by a green line in Fig 4a and 4b. In Figs. 6a and 6b, in accordance with request 13), DL plots are shown in red and LQ7 in black. The marker and font sizes have been increased for more legibility.

11) The black dashed line indicating the proxy for the CBL top height in Figure 5 is a little difficult to see. Could the line be made thicker?

The dashed is now plotted in bold. We also expanded the font size in Figs. 5a and 5b for more legibility.

12) The thin red lines in right hand plots of Figure 9 showing the availability of LQ7 data (and to some extent the thin blue lines showing the availability of DL data) are initially difficult to see. Creating separate plots for these lines could help.

We understand that the plot showing data availability may not be immediately clear because it is included in the right panel showing the mean profiles. However, including a third panel would require reducing the size of the contour plot, which contains the most important information. To preserve the clarity and visibility of the contour plot, we opted to present the data availability within the existing panel, even though it may require a bit more effort to interpret. The data availability profiles are now plotted in bold dashed lines and the colors have been modified (red: DL, black: LQ7) to fit the colors of the other figures.

13) It would be useful to consistently use the same colours to represent DL and LQ7 data in Figures 5, 6, 7, 9 (right hand plots), and 13. Moreover, it would be helpful to use these colours for the vertical lines in Figure 6 to distinguish between the scale sizes used for the different instruments.

In the revised version, DL plots are shown in red and LQ7 in black in Figs 5, 6 7, 9 and 13. In Fig. 8, the averaged curve has been replotted in blue (instead of red) to prevent misinterpretation, as red is used for all DL information.

TECHNICAL CORRECTIONS

14) The name O'Connor, in the reference to O'Conor et al. (2010) should be shown with the letter "c" in upper case rather than in lower case.

corrected

15) The DOI https://doi.org/10.5194/amt-2023-38 for Luce et al. (2023a) shown in the manuscript refers to the discussion version of the paper. The DOI https://doi.org/10.5194/amt-16-3561-2023, which refers to the accepted version, should be used instead.

corrected

---

## Author Comment (AC2)

Responses to reviewer 1
General comments and recommendation for disposition of the paper

This work presents an analysis on the observations made jointly by a Doppler lidar (DL) and a UHF band wind profiler radar (LQ7) in convective boundary layers (CBL) in clear air. The database was built during a measurement campaign lasting about one month. The research mainly uses the vertical profiles of the time series of the vertical air velocity W measured continuously by the two instruments. The information on the height Zi of the atmospheric boundary layer is deduced from the vertical reflectivity profiles of the LQ7 (local maximum of reflectivity). This important information is used in the discussions. As the title indicates, the authors focused the results on ε which represents the dissipation rate of the turbulent kinetic energy (TKE). It is a fundamental parameter in the description of the boundary layer. Given the role that W plays in the restitution of ε and in the demonstrations related to the inter-instrument comparison, it would be appropriate to include the vertical velocity in the title.

The difficulty of the comparison comes from the fact that the measurements of the two instruments do not overlap in altitude. The lidar produces observations below 300 m while the radar begins its usable observations from 400 m. The analysis then calls upon considerations of vertical continuity of W and ε and the known vertical evolution of ε in the CBLs. A true quantitative comparison remains to be made but from a qualitative point of view the results obtained are very convincing. The dissipation rate of the turbulent kinetic energy ε is obtained from W in two different ways. The first method (TS method) deduces ε from the spectral analysis of the time series of W under the assumption of Kolmogorov-Obukov inertial turbulence. The second method (DS method) which is the most commonly used in the literature uses the spectral variance $\sigma_t^2$ of the Doppler spectral peaks caused by turbulence. This information is only known and recorded by the radar. As expected due to the low temporal resolution of the radar (more than 10 times lower than DL), the TS method applied to radar data turns out to be unreliable in comparison with DL retrieval. In fact the comparison of the ε obtained by DL with the TS method and by LQ7 with the DS method is the main objective of the work. The various results of this comparison show a good similarity between these two types of measurement. These results are optimized by using for LQ7 the law $\varepsilon = \sigma_t^3 / (\alpha D)$. Where D is the thickness of the CBL and α is a parameter, comprised between 0.1 and 0.2, which depends on the physics of turbulence. This law, the formulation of which is found in the literature, remains empirical and needs further observations to be validated and in particular to quantify its uncertainty range.

The measurement of the lidar and radar has a spatial resolution that depends on the resolution volume as a function of the beam aperture and the pulse length. This induces a spatial filtering whose effect is mentioned but which is not taken into account here based on a rather weak justification. This filtering is taken into account in retrieval equations that can be found in the literature. The resolution volume is a parameter that differs greatly between the two instruments. It would therefore be important to evaluate the uncertainty brought by ignoring this filtering.

This well-structured document written in a clear and concise style is easy to read and understand. The demonstrations are supported by well-chosen graphs. The bibliography, which seems well up to date on the subject, shows that the authors are recognized specialists in the field covered. This work certainly convinces the reader of the interest of using a radar or a lidar for the measurement of the rate of dissipation of turbulent kinetic energy ε in the monitoring of CBLs in connection for example with atmospheric pollution problems.

In conclusion this paper which is well conducted and which presents a good scientific interest is appropriate for the publication in the journal Atmospheric Measurement Technique. However this recommendation is subordinated by the consideration by the authors of the previous major and following minor comments.

We thank the reviewer for the evaluation of our manuscript and for his positive assessment. We understand that the referee's primary concern is our lack of discussion regarding the effect of radar spatial filtering on the DS method. This is indeed a crucial aspect, which we address here in greater detail and have incorporated into the revised version of the manuscript.

Main review points

1) The technical description of the instruments in the document is brief, it is supplemented by bibliographical references. In order not to tire the reader too much in often tedious research, I suggest giving directly the essential operating parameters of the instruments: length and coding of the pulses, number, direction, and aperture of the beams used.... This would allow to quickly have an idea on the spatial resolution and to understand why the LQ7 has a first gate starting at 300 m while the given vertical resolution is 100 m.

We added a table with the specification of the two instruments.

| | LQ7 | DL |
|---|---|---|
| Operational frequency | 1.375 GHz | 194 THz (Wavelength: 1.543 µm) |
| 2 ways half power half width (°) | 2.1 | |
| Beam directions (Az,Ze) (°) | (0,0), (0,14.2), (90,14.2), (180,14.2), (270, 14.2) | (0,0), (0,28), (90,28), (180,28), (270, 28) |
| Range resolution (m) | 100 | 20 |
| Number of gates | 80 | < 20 |
| Altitude of the first gate (AGL) | 300 | 40 |
| Interpulse Period ($\mu$s) | 100 | 0.33 (Shot frequency: 3000 Hz) |
| Acquisition time for one profile (s) | 59 | 4 |
| Acquisition time of the mean profile for routine measurements (min) | 10 | 10 |
| Velocity aliasing ($ms^{-1}$) | 10.8 | 44 |

Table 1: Main specifications of LQ7 and DL

2) The DS method uses the variance of the spectral Doppler peaks corrected for effects not related to atmospheric turbulence. It is important to list all of these extra-turbulent causes and to explain how the correction is made.

Following the suggestions of both reviewers, the methodology for applying the spectral width technique has been expanded by including an appendix.

3) Figure 1 shows that in addition to the vertical beam, the lidar has four oblique beams and the radar five. The authors have used only the vertical beam for the direct measurement of W. We can also use only the oblique beams to deduce W. When the components of the horizontal wind speed are far from the zero spectral line, the measurement of W will be less sensitive to the presence of ground echoes which are very penalizing at low levels (mainly for the radar). I would like to know if the authors have used this possibility and what advantages and disadvantages they found in it.

For the present study, we did not use the oblique beams to deduce *W*. We note the two independent estimates as follows:

$$W_{NS} = \frac{RN + RS}{2\cos(\alpha)}$$
$$W_{EW} = \frac{RE + RW}{2\cos(\alpha)}$$

Where $\alpha = 28°$ for DL and $14.2°$ for LQ7, and RN, RS, RE and RW are the radial velocities from North, South, East and West directions, respectively.

The first author considered this possibility with the MU radar for a climatology of frequency spectra in a recent paper (Luce, H., Nishi, N., & Hashiguchi, H. (2024). A climatological study of the frequency spectra of vertical winds from MU radar data (1987–2022). Journal of Geophysical Research: Atmospheres, 129, e2024JD041677. https://doi.org/10.1029/2024JD041677).

The wind speed near the surface remained consistently weak throughout the entire campaign, with values below a few meters per second, except for 06 September 2022, which is excluded from the statistical analysis. Comparisons between $W$, $W_{NS}$, and $W_{EW}$ showed no significant differences across the dataset used in the manuscript. This suggests that ground clutter contamination was likely negligible in LQ7 data. In response to the reviewer's request, we present several figures comparing the three velocity estimates from LQ7 and DL and the corresponding spectra for the example days highlighted in the manuscript, specifically 27 August and 11 September 2022.

Figures R1a and R1b show the time series at 3-min time resolution of $W$, $W_{NS}$, and $W_{EW}$ spanning from 07:00 LT to 17:00 LT on 27 Aug and 11 Sep , for LQ7 in the range 400-500 m and DL in the range 100-300 m. Figures R2a and R2b show the corresponding scatter plots of $W$ vs $W_{NS}$ (left panel), $W$ vs $W_{EW}$ (middle panel), $W_{NS}$ vs $W_{EW}$ (right panel) for DL (red) and LQ7 (black). The time series reveal no substantial outliers or biases. Scatter plots for DL and LQ7 on 11 September indicate greater dispersion for both instruments. This increased dispersion suggests that it is not solely attributable to estimation errors but also reflects real differences between the estimates, likely caused by the sampling of different volumes. The mean difference between $W$, $W_{NS}$, and $W_{EW}$ varies between 0 to 4 cm/s for both instruments.

[Figure]

Figure R1a: Time series of vertical velocities W, $W_{NS}$ and $W_{EW}$ at a time resolution of 3 min between 07:00 LT and 17:00 LT on 11-SEP-2022 for DL (top) [100-300 m] and LQ7 (bottom) [400-500 m].

[Figure]

Figure R1b: Same as Fig R1a for 27-AUG-2022.

[Figure]

Figure R2a: Scatter plots of W vs $W_{NS}$ (right), W vs $W_{EW}$ (middle) and $W_{EW}$ vs $W_{NS}$ for the data shown in Fig. R1 on 11-SEP-2022 for DL (red) and LQ7 (black).

[Figure]

Figure R2b: Same as Fig R2a for 27-AUG-2022.

Figures R3a and R3b show the corresponding wavenumber or frequency spectra (blue from $W$, green from $W_{NS}$, and red from $W_{EW}$). They globally show the same shape at all frequencies or scales for both DL and LQ7. However, we have noted that the spectral levels of $W_{NS}$ and $W_{EW}$ are slightly lower than those of W for DL (on the order of 30% to 50% , Fig. R3b). The variances of the $W_{NS}$ and $W_{EW}$ time series from DL are also lower than the variance of W (Fig. R1a and R1b).

[Figure]

Figure R3a: Wavenumber spectra of W (blue), $W_{NS}$ (green) and $W_{EW}$ (red) from LQ7 data for 11-SEP-2022 and 27-AUG-2022.

[Figure]

Figure R3b: Frequency spectra of W (blue), $W_{NS}$ (green) and $W_{EW}$ (red) from DL data for 11-SEP-2022 (left ) and 27-AUG-2022 (right).

We suspect that this property may result from the sampling of different volumes in a horizontally inhomogeneous W field. This effect is likely more pronounced for DL due to its oblique beams being tilted 28° off zenith and its very narrow beam. Within the 100–300 m range, the horizontal distance between the lidar sampling volumes is approximately 100–300 m. Note that for LQ7, the corresponding horizontal distance within the 400–500 m range is approximately 110 m only (tilt angle of the beams=14.2°)

We made an elementary test by assuming a wave disturbance anomaly of W (to simulate updrafts and downdrafts within a CBL) of amplitude 1 m/s, with a period of 100 s and a horizontal wavelength of 1000 m at z=200 m. We assimilate the DL as a point measurement and we generated W time series from the vertical beam and the combination of the two pairs of oblique beams tilted 28° off zenith. The objective is not to produce realistic values but to get a qualitative result. The time series of measured and reconstructed W are shown in red and blue curves, respectively (Fig. R4). The reconstructed W amplitude is indeed attenuated by 20%. Therefore, $W_{NS}$ and $W_{EW}$ may be underestimated when the horizontal distance between the pairs of DL sampling volumes is not negligible compared to the horizontal scale of the W disturbance. This may represent a significant drawback when estimating

energy parameters from $W_{NS}$ and $W_{EW}$. More realistic simulations, coupled with a quantitative analysis of the time series, could provide valuable insights into the horizontal inhomogeneity of the vertical wind field.

[Figure]

Figure R4: Schematic representation of the impact of the use of 2 oblique beams on W measurements

4) The study offers a database of vertical air speed observed by two different types of instruments of rare quality but which I find insufficiently exploited. This concerns in particular the problem of the negative bias of the average vertical speed in the CBL observed by the UHF band wind profiler radar since their beginning in the seventies. This negative bias can reach several -0.1 ms-1. Experimental and theoretical explanations have been put forward but to my knowledge without success and the problem remains. On the contrary in your work you deduce that this bias is almost zero for the UHF radar and also for the lidar. It is frustrating for an informed reader that you do not point out this contradiction with the numerous past observations. This paper would therefore benefit if you developed this particular point even if it does not fit into the main objective of the paper.

The reviewer seems to refer to results reported by e.g., Angevine (1997) and references therein. These results show a wind profiler produces a negative W bias (for unclear reasons) during CBL activity.

The analysis we presented in the manuscript does not allow us to confirm or rule out the presence of such a bias in our dataset. It is limited to the daytime, partly because of birds or bats contaminations of the raw data during night time. If the bias is present, it would not affect our spectral analysis (because it has a diurnal time scale). For information only, we plotted the histograms of W(DL, 100-300 m) and W(LQ7, 400-500 m) measured between 07:00 LT and 17:00 LT for the whole period of the campaign (27 Aug – 15 Sep) in Fig. R4. The mean difference <W(LQ7)-W(DL)> is -10 cm/s. It is consistent with the possibility of a bias in LQ7 measurements, assuming that DL measurements are unbiased. (NB: Angevine (1997) suggested that radar is partly sensitive to suspended particles to explain the measurement bias. Lidar would undoubtedly be affected by the same bias, and comparisons would be unable to identify it.) Unfortunately, this difference must be interpreted with caution, as the two instruments do not sample in the same altitude range. Like the results we present in this manuscript, data covering the same altitude range should be obtained for more decisive analysis and conclusions.

[Figure]

Figure R5: Histograms of W from LQ7 [400-500 m] and DL [100-300 m]using all the data from 27-AUG to 15-SEP-2022.

5) The localization of the height of the boundary layer Zi by a local maximum of the radar reflectivity is a method which if well conducted gives precise results. It is therefore not necessary to associate the word proxy with this radar Zi which in fact tends to discredit this measurement.

We removed the term "proxy".

6) The observations that are discussed are located in the convective boundary layer. This is a dynamical and thermodynamic framework that is inefficient for the development and maintenance of gravity waves. It is therefore unwise to dwell on it too much.

The reviewer is right. We address this point in lines 465-466.

7) For figure 12 use $\log10\ (\varepsilon\ DSLQ7\ /\ \varepsilon DL)$ instead of $\log10\ (\varepsilon DSLQ7)- \log10\ (\varepsilon DL)$ in the caption and instead of $\Delta \log10\ (\varepsilon)$ on the horizontal axis.

We corrected the caption and label of figure 12

---

## Author Response (AR2)

**Responses to the reviewer 2  (2nd review)**

This is the 2nd time that I have reviewed this submission. I didn't have any substantial comments on the previous draft and my feedback mostly related to technical issues. I note that these have been addressed appropriately by changes made to this version of the manuscript.

We warmly thank the reviewer for the detailed revision of the manuscript and for the time he dedicated to it. His valuable input has significantly contributed to the quality of the paper, and we express our gratitude in the 'Acknowledgements' section.

I have raised a number of issues with the revised manuscript, but they are all minor technical issues.

1) Line 66: The meaning of the symbol D (i.e. the depth of CBL) should be given here since it was previously only given in the abstract. (I note that the same symbol is used with a slightly different meaning, i.e. the depth of shear-generated turbulent layer, on line 153 although I have no problem with that).

The parameter D plays a central role in this paper; therefore, we have adjusted the text to consistently use the symbol throughout  (and in Figures 8 and 11). It is now introduced in Lines 50-53 as follows:

"Reliable and continuous measurements of this parameter covering the whole CBL depth (*hereafter denoted as D throughout the paper*) are necessary to assess and improve the subgrid turbulence schemes in numerical weather forecast models."

We retained the same notation for the depth of the shear-generated turbulent layer (line 153) and revised the text to clarify that this choice was intentional. After Eq. (3): "Here, $D$ represents the depth of the shear-generated turbulent layer"

2) lines 68 and 130. The reference to O'Connor et al. (2010) has been corrected on line 550 of the References section, i.e. by showing the letter "C" in upper case. However, it is still shown with a lower case "c" within the main body of the text.

Corrected

3) line 81: The words "Very High Frequency" should be moved to line 71, where the abbreviation VHF is first used.

Indeed. It is correctd

4) line 168: "Outside this time window, the two instruments do not sample a priori the same atmospheric dynamics due to the absence of height overlaps." There is an absence of height overlaps at all times, so should this sentence instead be referring to the fact that the two instruments are not both sampling the CBL outside of this time window?

The reviewer is right. We have modified the sentence as follows: "Outside this time window, the CBL is not sampled by the two instruments."

5) line 176: Reference to "Figure 3a" should be to "Figure 4a".

Corrected: "Figure 3a" -> "Figure 5a"

6) Figure 2: It would be better to swap the order of the two panels so that the plot for 11th September is shown first as in other figures. The caption for Figure 2 suggests that this was the authors' intention.

Done.

7) line 200: "The cloud base is highlighted by black dots". These dots are not clearly visible, in part because they are superimposed on dark red colours. Maybe the authors could try using white dots instead?

The cloud base is now marked by white dots, providing a clearer indication of its location.

8) line 204: "The height of the cloud base is consistent with the height of the lifting condensation level (blue curve) calculated from surface meteorological data collected at the observatory using Emanuel's (1994, pp 129- 130) derivations." The LCL curve is green rather than blue. The same mistake is made in the caption for Figure 4.

Corrected.

9) Figure 4: The caption should explain the the meaning of the vertical lines, i.e. indicating the 07 - 17 LT time window.

The caption now indicates: "The vertical solid black lines show the [07:00-17:00] LT time window used for the comparisons between LQ7 and DL. "

10) line 278: "The red dashed line shows the theoretical inertial spectrum (−5/3 slope) and the blue dotted curve, the best fit of the DL spectrum with the theoretical Kristensen et al. (1989) 1-D line spectrum using $\mu$ = 1". There are two types of red dashed lines in Figure 6a. I think the authors are referring here to the -5/3 slope line rather than to the vertical red dashed lines? Also, I think that the authors intended to refer to the "blue dashed line" rather than to the "blue dotted curve".

We are sorry for the confusion. The text referred to the red dotted curve (not blue) showing the best fit of the DL spectrum with the theoretical *Kristensen et al. (1989)* 1-D line spectrum. It is now corrected in the text and in the caption.

11) line 283: "Even if the calculated slopes are flatter than −5/3, the two spectra are not inconsistent with an inertial subrange as suggested by the red dashed line." As for point 10, the authors should clarify which red dashed line they are referring to, i.e. the one for the -5/3 slope.

‘Red dashed line’ should be corrected to ‘Blue dashed line,’ as it was mistakenly carried over due to color changes between the submitted and revised versions. Several similar inconsistencies (as reported in 15 and 16 by the reviewer) were found throughout the text and have now been corrected.

12) line 284: "The LQ7 spectrum agrees well with the high wavenumber part of the DL spectrum in slope, shape and level suggesting that the same turbulent regime was detected by both instruments." I think the authors intended to refer to the low (not high) wavenumber part of the spectrum?

Yes, it is corrected.

13) line 286: "Note that the two wavenumber ranges of the spectra overlap more than those shown in Fig. 6a because the wind speed was ~2.5 m s−1 in both altitude ranges selected for DL and LQ7." The two wavenumber ranges do not overlap, but are closer together.

We revised the wording as per the reviewer's suggestion.

14) line 291: " Incidentally, the LQ7 spectrum exhibits a peak at $k_m$~0.07 $rad\ m^{-1}$, a −5/3 slope at higher wavenumbers and a flat spectrum at lower wavenumbers down to 2 $10^{-3}\ rad\ m^{-1}$" Should the reference to "0.07 rad m-1" instead be t .007 rad m-1", i.e. to 0.7 10-2 rad m-1"?

Yes. It is corrected.

15) line 320: "(i.e. the differences between the black and blue curves in Fig. 7)". I think that the authors intended to refer to the "red" rather than "blue" curve?

Yes, it is corrected

16) line 323: "The red curve shows . . ." I think that the authors intended to refer to the "blue" rather than "red" curve in Figure 8?

Yes, it is corrected.

17) line 447: "Due to the lack of range overlap, comparisons make sense for convective boundary layers only, for which we expect some degree of homogeneity with height in the

mixed layer." I think the authors need to refer to "well-developed convective boundary layers" rather than just "convective boundary layers" here?

"well developed is added . The term indeed refers to mature and a quasi-steady-state structure with a well-mixed layer, likely a necessary condition for good agreements between the instruments.

18) lines 466, 470, and 473: The authors have started to use the subscript "t" for "$\varepsilon$" in these lines, but not elsewhere in the manuscript. Is this intentional ?

It is not intentional. The subscripts have been removed.

19) line 490: "The finite radar volume (noted 2a and 2b in the radial and transverse directions, e.g., Hocking et al., 2016) and the limited acquisition time (temporal resolution) $\Delta T$ can play the role of spatial filters if the outer scales of turbulence $Lout$ are much larger than 2a, 2b and $U\Delta T$." I am noting that the meanings of symbols "a" and "b" have not be given in the appendix. The authors should consider whether the meanings should be given.

The meaning of the symbols is now given.